# Model-based design of a wave-feedforward control strategy in floating wind turbines

Alessandro Fontanella[1], Mees Al[2], Jan-Willem van Wingerden[3], and Marco Belloli[1]

[1]Mechanical Engineering Department, Politecnico di Milano, Milano, Via La Masa 1, 20156, Italy.
[2]Sowento GmbH, Donizettistraat 1A, 70195 Stuttgart, Germany.
[3]Delft Center for Systems and Control, Delft University of Technology, Delft, 2628 CD, The Netherlands.

**Correspondence:** Alessandro Fontanella (alessandro.fontanella@polimi.it)

**Abstract.** Floating wind turbines rely on feedback-only control strategies to mitigate the negative effects of wave excitation. Improved power generation and lower fatigue loads can be achieved by including information about incoming waves into the turbine controller. In this paper, a wave-feedforward control strategy is developed and implemented in a 10MW floating wind turbine. A linear model of the floating wind turbine is established and utilized to understand how wave excitation affects rotor-speed, and so power, as well as to show that collective-pitch is suitable for reducing the effects of wave excitation. A feedforward controller is designed based on the inversion of the linear model, and a gain-scheduling algorithm is proposed to adapt the feedforward action as wind speed changes. The performance of the novel wave-feedforward controller is examined first by means of linear analysis, and then with non-linear time-domain simulations in FAST. This paper proves that including some information about incoming waves into the turbine controller can play a crucial role in improving power quality and the turbine fatigue life. In particular, the proposed wave-feedforward control strategy achieves this goal complementing the industry-standard feedback pitch controller. Together with the wave-feedforward control strategy, this paper provides some insights about the response of floating wind turbines respond to collective pitch control and waves, that can be useful in future control-design studies.

## 1 Introduction

Floating offshore wind turbines (FOWTs) are currently operated without any real-time information about ocean conditions. Industry-standard controllers are feedback (FB) only: the wind turbine controller reacts to the external disturbance of wind and waves as this occurs. One possibility to improve the current floating wind technology is to include real-time information about the marine environment into the turbine controller and to design new control logics based on that.

Concerning wind turbulence, feedforward (FF) control has recently drawn the attention of the research community, as it can effectively reduce fatigue loads and improve power production. Research has been mainly driven by improvements in the LIDAR (light detection and ranging) technology that enables measurement of the wind field upstream the wind turbine. One of the first studies about LIDAR-assisted control was carried out by Harris et al. (2006), that explored the potentialities of this new control strategy, and found it can reduce blade fatigue loads approximately of 10% in turbulent wind conditions. Since then, several control logics were developed based on the inclusion of LIDAR signals in the turbine controller, for example

by Laks et al.; Dunne et al. (a,b,c); Schlipf et al. (2013), demonstrating positive effects for the blade loads and the turbine components. Very few are the studies about LIDAR-assisted wind-FF control in floating wind turbines: in the paper of Schlipf et al. (2015), a collective-pitch FF controller is designed to reduce rotor speed oscillations caused by inflow turbulence. The LIDAR moves with the floating turbine and measurements need to be corrected for that. The proposed FB-FF controller, that keeps into account the above-mentioned movement of the LIDAR, can reduce power and rotor speed fluctuations up to 80% and tower, rotor-shaft, and blades fatigue loads of 20%, 7% and 9%, respectively.

Wave disturbance is responsible of a considerable fraction of dynamic excitation experienced by an FOWT. This was first shown in the work of Jonkman (2007), where the response of NREL 5-MW turbine installed on land is compared to the same turbine mounted on the floating ITI Energy barge, in the presence of wind and waves. The rotor-speed excursions in the floating turbine are increased of 60% because of the oscillations in wind speed caused by platform motion. Consequently, fluctuations in the generator-power are larger as well. Tower shear forces and bending moments are increased. The analysis of Jonkman (2007) also shows that offshore-to-onshore ratios decrease with decreasing severity of the wave conditions, suggesting that a large part of the increments is due to wave excitation. More specifically for the drivetrain, Nejad et al. (2015) assessed the loads in a 5MW wind turbine mounted on four platform concepts for different wind and wave conditions. The analysis suggests an increment of the fatigue damage, which is mainly caused by the large wave-induced thrust force.

Applying the same idea behind wind-FF control to wave is therefore an attractive perspective, but the idea is largely unexplored. Raach et al. (2014) introduced an NMPC strategy that uses a perfect preview of a reduced wave disturbance to mitigate the turbine structural loads. Promising improvements over an industry-standard FB controller are shown, at the expense of a significant increase in the controller complexity. Moreover, most of the performance gain is seen for blade loads, that are caused by wind turbulence rather than wave, so it appears the NMPC does not effectively counteract waves. Ma et al. (2018) developed and validated two algorithms for real-time forecasting of wave forces. Based on the predicted wave forces, a finite-horizon LQR controller is designed and applied to a TLP-FOWT to minimize the tower-base fore-aft bending moment, achieving mixed results. Al et al. (2020) introduced an inversion-based feedforward control strategy, showing it is an effective way of reducing wave-induced rotor speed oscillations.

The present paper further develops the concept of wave-FF exploiting tools of model-based control. The wave-FF control strategy is enabled by an integrated model of the FOWT that captures its most relevant physics. Hence, this work proves the effectiveness of multidisciplinary analysis as a mean to advance the current floating wind technology.

All the reasoning is made with reference to a floating wind turbine, but it is deemed valid for any FOWT. The floating wind turbine of reference is based on an open-source concept and is defined in Section 2. The idea is to use tools of multivariable systems control to gain insight about the effects of waves on the FOWT response and assess which is the best control input (generator torque or collective pitch) to mitigate them. Then, to leverage this knowledge to design a feedforward controller that reduces power fluctuations caused by waves. A control-oriented linear model of the FOWT is required first for the multivariable analysis, and later for the synthesis of the feedforward controller. The control-oriented linear model is briefly introduced in Section 3. Section 4 deals with the input-output analysis. The feedforward controller is designed in Section 5. Again, linear analysis is utilized to assess the controller performance, which is shown to be highly dependent on the wind turbine operating point.

Hence, a gain-scheduling law is introduced to have the maximum performance in any wind condition. The wave-feedforward controller requires as input a preview of the incoming waves, which is obtained based on the algorithm presented in Section 6. In Section 7, the feedforward controller and the wave prediction algorithm are implemented in a nonlinear, medium-fidelity model of the floating wind turbine, and numerical simulations are carried out in realistic environmental conditions to evaluate the benefits of the feedforward control strategy. Section 8 draws the conclusion and gives some recommendations for future work.

## 2 Definition of a reference floating wind turbine

This section defines the floating system that is considered in this study. The FOWT is formed by the DTU 10MW (Bak et al. (2013)) wind turbine and the INNWIND.EU TripleSpar platform (Azcona et al. (2017); Lemmer et al. (2020a)). The characteristics of this FOWT concept are similar to those of the current commercial projects and are publicly available.

The floating wind turbine is regulated with an industry-standard generator-speed controller. In below-rated winds the controller maximizes the extracted power by keeping the blade pitch angle $\theta$ constant and varying the generator torque $Q_G$ as a function of generator speed $\omega_G$ squared:

$$Q_G = k_G \omega_G^2 \,, \tag{1}$$

with $k_G = \frac{1}{2}\rho\pi R^5\left(C_{p,max}/\tau^3\lambda_{opt}^3\right)$, where $\rho$ is the air density, $R$ the rotor radius, and $\tau$ the transmission ratio. $C_{p,max}$ is the maximum power coefficient, which is achieved for zero pitch angle and the optimal tip-speed-ratio $\lambda_{opt}$.

In above-rated winds, the controller regulates the extracted power to its rated value setting the generator torque to a constant value, equal to rated. Generator speed oscillations are directly reflected by the wind turbine power output. Rotor speed is regulated to its rated value $\omega_{G,r}$ by the collective-pitch controller (CPC), which reacts to the generator speed feedback as:

$$\theta = k_P(\omega_G - \omega_{G,r}) + k_I \int (\omega_G - \omega_{G,r})dt \,, \tag{2}$$

where $k_P$ and $k_I$ are the proportional and integral gains, tuned following the model-based approach of Fontanella et al. (2018) to achieve the maximum damping for the platform pitch mode and for the drivetrain mode. A gain scheduling factor is introduced to adjust the PI controller gains as wind speed varies. The generator-speed feedback controller constitutes the baseline configuration against which the benefits of wave-FF are assessed.

## 3 The control-design model

The wave-FF control strategy we want to develop is model-based, and its development requires a linear model of the floating wind turbine. The control-design model is derived based on linear first-principle equations of the most important physics of the FOWT, rather than from the linearization of a higher-order model. The main features of the model are recalled below, while a detailed description is reported in the article of Fontanella et al. (2020). The model describes the global dynamics of the FOWT,

neglecting the dynamics of single components. It considers the rigid-body platform motions and the rotor dynamics about a steady-state configuration (operating-point) set by an average wind speed, platform motion, rotor speed, collective-pitch and generator torque. The inputs are generator torque and collective blade-pitch angle, which are the main control variables for the FOWT, in addition to wind turbulence and wave elevation. The model equations are cast in state-space form and are valid only for small perturbations about the operating-point.

The structural dynamics builds on the theory of multibody systems. The model considers the FOWT components as rigid bodies: this simplification is deemed acceptable in a control-oriented model, as the bandwidth of an FOWT controller is usually lower than the flexible modes of the tower, blades, and drivetrain. Moreover, the focus of the control-oriented model is the coupled rotor-platform response induced by waves more than the dynamics of the flexible components.

Rotor aerodynamics are introduced in the model with a simplified approach. The aerodynamic model does not consider the single blade but computes the integral rotor forces. This simplification is valid because the FOWT global dynamics is determined by the integral rotor loads, rather than the loads of the single blades. This assumption is reasonable for a reduced-order model of the FOWT, as noticed by Lemmer et al. (2020b). Only the rotor torque and thrust force are considered because they drive the global dynamics of the floating turbine: aerodynamic torque sets the wind turbine power production and thrust force the motion of the floating platform. Torque and thrust are modeled by means of the quasi-steady approach, based on the derivatives of the torque and thrust curves of the wind turbine. The formulation of the control-oriented model enables the inclusion of unsteady aerodynamic effects associated with the FOWT motion, which may have an influence on the platform response. In this respect, a similar approach to the one presented by Bayati et al. (2017) could be used.

### 3.1 Frequency-dependent hydrodynamic loads

Hydrodynamic radiation and first-order-wave forces are modeled by means of linear-time-invariant parametric models.

Frequency-dependent radiation forces are approximated by a parametric model in state-space form, from the added mass and damping matrices of panel code pre-calculations. In this work, the frequency domain identification method of the MATLAB toolbox developed by Perez and Fossen (2009) is used, but other methods are available in literature, for example the one by Janssen et al. (2014).

Also the first-order-wave excitation is introduced in the model with a parametric model in state-space form, which connects the wave elevation to the wave forces. This choice allows to have the wave elevation, rather than wave forces, as input to the model. The wave excitation model is obtained based on the wave-force coefficients, which are usually computed at discrete frequencies through a panel code (e.g. WAMIT). In the present case, the parametric model is defined by means of system identification of the impulse response function of the force coefficients. This approach was firstly applied to floating turbines by Lemmer et al. (2020c), and it is currently available to model wave excitation in OpenFAST (Jonkman et al. (2018)). Identification of the parametric model from frequency-domain data (i.e. the force coefficients) is also a possibility, that was for example used by Al et al. (2020). The wave-force model of panel code data is non-causal, which means a force is developed before the wave reaches the center of the platform. The panel code data, in the form of impulse response functions, are causalized before system identification by introducing a time delay $t_d$ (with $t_d > 0$) to have zero response for negative times.

The time delay is embedded inside the identified parametric model and, consequently, into the linear model of the FOWT. The response obtained from the linear model is delayed of a time $t_d$ with respect to the input wave elevation.

## 4 FOWT response to controls, wind and waves

An input-output analysis is carried out to gain insight into the FOWT response to the available controls, generator torque and collective-pitch, and to the wave disturbance. The analysis answers the question of which is the best combinations of controls to reject the negative effects of waves. This information is used later to support the synthesis of the wave-FF control strategy. Moreover, the analysis gives a picture of the FOWT dynamics that may prove to be useful also for other purposes.

The analysis starts from the control-design model in a transfer function representation

$$\boldsymbol{y} = \boldsymbol{G}\boldsymbol{u} + \boldsymbol{G}_d\boldsymbol{d}. \tag{3}$$

The system has two outputs, rotor speed and tower-top motion, collected in $\hat{\boldsymbol{y}} = [\omega_r, \ x_{tt}]^T$; two control inputs, collective-pitch and generator torque, collected in $\hat{\boldsymbol{u}} = [\theta, \ Q_g]^T$; and two disturbance inputs, variation from average of the hub-height wind speed and wave elevation, $\hat{\boldsymbol{d}} = [v, \ \eta]^T$. The model of Eq. (3) is used to compute the outputs deviation from their steady-state value, due to a change in the control and disturbance inputs.

To facilitate the interoperation of MIMO analysis results, the model of Eq. (3) needs to be scaled. This ensures that inputs and outputs are of the same importance. The scaled model is obtained by dividing any variable by its maximum expected (for disturbances) or allowed (for control inputs) change. The output, input and disturbance scaling matrices ($D_y$, $D_u$ and $D_d$ respectively) are

$$\boldsymbol{D}_y = \begin{bmatrix} 0.15\omega_0 & 0 \\ 0 & 5 \end{bmatrix}, \qquad \boldsymbol{D}_u = \begin{bmatrix} 5\pi/180 & 0 \\ 0 & 0.1Q_{g,0} \end{bmatrix}, \qquad \boldsymbol{D}_d = \begin{bmatrix} 0.1U & 0 \\ 0 & 4 \end{bmatrix}, \tag{4}$$

where $\omega_0$ is the rated rotor speed, $Q_{g,0}$ the rated generator torque, and $U$ the mean wind speed. The scaled model is

$$\hat{\boldsymbol{y}} = \hat{\boldsymbol{G}}\hat{\boldsymbol{u}} + \hat{\boldsymbol{G}}_d\hat{\boldsymbol{d}}, \tag{5}$$

with

$$\hat{\boldsymbol{G}} = \boldsymbol{D}_y^{-1}\boldsymbol{G}\boldsymbol{D}_u, \qquad \hat{\boldsymbol{G}}_d = \boldsymbol{D}_y^{-1}\boldsymbol{G}_d\boldsymbol{D}_d. \tag{6}$$

Given an input between 0 and 1, where 0 is no input and 1 is the maximum expected value, the outputs of the model of Eq. (5) take a value between 0 and 1, where 0 is no output and 1 corresponds to the maximum expected or allowed value for the output.

### 4.1 Control inputs

The model without disturbances $\hat{\boldsymbol{y}} = \hat{\boldsymbol{G}}\hat{\boldsymbol{u}}$ is considered first. The transfer function matrix $\hat{\boldsymbol{G}}$ has two couples of input and output directions, each with an associated gain. For any selected frequency, the directions and gains of matrix $\hat{\boldsymbol{G}}$ are obtained

from its singular value decomposition (SVD) (Levine (1996))

$$\hat{\boldsymbol{G}} = \boldsymbol{U}\boldsymbol{\Sigma}\boldsymbol{V}^H. \tag{7}$$

The column vectors of $\boldsymbol{V} = [\overline{\boldsymbol{v}},\ \underline{\boldsymbol{v}}]$ are the input directions, the column vectors of $\boldsymbol{U} = [\overline{\boldsymbol{u}},\ \underline{\boldsymbol{u}}]$ the output directions, and the respective singular values are along the diagonal of $\boldsymbol{\Sigma} = \mathrm{diag}(\overline{\sigma},\ \underline{\sigma})$. When the input vector $\hat{\boldsymbol{u}}$ has the same direction of vector $\overline{\boldsymbol{v}}$, the output $\hat{\boldsymbol{y}}$ is along the direction $\overline{\boldsymbol{u}}$, the gain is equal to $\overline{\sigma}$ and it is the largest possible for that frequency. The input produces the most effect on the output, and the directions of $\overline{\boldsymbol{v}}$ and $\overline{\boldsymbol{u}}$ are named the *strongest*. Conversely, when the input is directed as $\underline{\boldsymbol{v}}$, the gain is $\underline{\sigma}$, and the input has the least effect on the output, which is along $\underline{\boldsymbol{u}}$. The directions of $\underline{\boldsymbol{v}}$ and $\underline{\boldsymbol{u}}$ are named the *weakest*.

The steady-state (i.e. zero frequency) plant model of the FOWT in a 16 m/s wind is

$$\hat{\boldsymbol{G}}(j\omega = 0) = \begin{bmatrix} -2.736 & -0.311 \\ -1.216 & 0.097 \end{bmatrix}. \tag{8}$$

The (1,1) element of $\hat{\boldsymbol{G}}$ is much larger than the (1,2) element, so rotor speed is a lot more sensitive to a steady-state (i.e. very slow) change in collective-pitch, the first input, than in generator torque, the second input. Collective-pitch has an effect both rotor speed, the first output, and tower-top motion, the second output, in the same direction. If collective-pitch is increased, the rotor is slowed because of the decreased aerodynamic torque, and the nacelle moves upwind, because of the lower thrust force. The plant model is decomposed into its SVD

$$\boldsymbol{U} = \begin{bmatrix} -0.916 & -0.401 \\ -0.401 & 0.916 \end{bmatrix}, \qquad \boldsymbol{\Sigma} = \begin{bmatrix} 3.004 & 0 \\ 0 & 0.214 \end{bmatrix}, \qquad \boldsymbol{V} = \begin{bmatrix} 0.997 & -0.082 \\ 0.082 & 0.997 \end{bmatrix}. \tag{9}$$

The strongest and weakest input directions are obtained by different combinations of collective-pitch and generator torque (e.g. the strongest is given by 0.997 of collective-pitch and 0.082 of generator torque). The ratio between the gain in the strongest and weakest direction (i.e. the condition number) is $\mathrm{CN} = \overline{\sigma}/\underline{\sigma} = 14.0$. The system is said to be *ill-conditioned*. At steady-state, the input combinations with prevailing collective-pitch have a much stronger effect on the FOWT than the input combinations with prevailing generator torque. The strongest and weakest output directions are given by different combinations of rotor speed and tower-top motion. From $\overline{\boldsymbol{u}} = [-0.916,\ -0.401]^T$, it is seen the effect of the strongest input combination, an increase of collective-pitch, is to slow down the rotor and to move the nacelle upwind. This is in agreement with the result of the inspection of $\hat{\boldsymbol{G}}(j\omega = 0)$ and makes sense from a physical point of view.

The SVD of the plant model $\hat{\boldsymbol{G}}(j\omega)$ is computed for several frequencies up to 0.3 Hz, and for seven operating points of wind speeds between 12 and 24 m/s. The top plot of Fig. 1 shows the magnitude of the two component of $\overline{\boldsymbol{u}}$, that is the fraction of collective-pitch and generator torque in the strongest input combination; the middle plot, the magnitude of the first and second component of $\overline{\boldsymbol{v}}$, the fraction of rotor speed and tower-top motion in the strongest output direction; the bottom plot the corresponding singular values, the gain. Collective-pitch is the most effective input, at any frequency and in any above-rated wind speed. Pitching blades affects both rotor speed and tower top motion, because it modifies at the same time the

aerodynamic torque and thrust. At the platform modes frequencies, the response is almost only tower-top motion, and the gain is increased: it takes a small collective-pitch action to move the nacelle, because the resulting rotor thrust variation excites the resonant response of the platform. Controlling rotor speed is hard. In the wave frequency range, the gain is decreased so it becomes more difficult to control the system, and rotor speed is easier to control than tower-top motion.

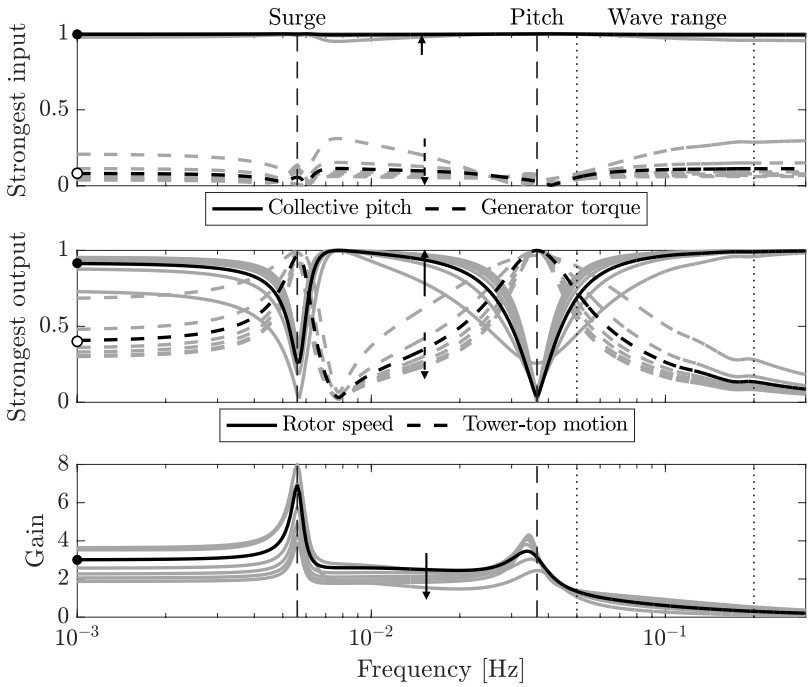

**Figure 1.** Singular value decomposition of the floating wind turbine plant for several above-rated operating points (grey, arrows for increasing wind), and for the 16 m/s wind case (black). Values for the zero-frequency case are displayed by the marks ● and ○. The vertical dashed lines are the frequency of the platform surge and pitch modes, the frequency range where waves are active is enclosed by the vertical dotted lines.

In summary, collective-pitch is the most effective control in above-rated winds. It has an effect both rotor speed and tower-top motion. In the frequency range where wave is active, collective-pitch becomes less effective, so it is harder to control the wind turbine.

### 4.2 Disturbances

The wind and waves disturbances are here considered separately. The direction of a disturbance is

$$190 \quad \boldsymbol{y}_d = \frac{1}{\|\hat{\boldsymbol{g}}_d\|_2}\hat{\boldsymbol{G}}_d\,, \tag{10}$$

where $\hat{\boldsymbol{g}}_d$ is the appropriate column of $\hat{\boldsymbol{G}}_d$ (the first for wind, the second for wave). The disturbance condition number is

$$\mathrm{DCN} = \overline{\sigma}(\hat{\boldsymbol{G}})\overline{\sigma}(\hat{\boldsymbol{G}}^{-1}\boldsymbol{y}_d)\,, \tag{11}$$

where $\overline{\sigma}(\cdot)$ is the maximum singular value. The DCN measures the control effort required to reject a given disturbance, relative to rejecting a disturbance with the same magnitude but aligned with the strongest output direction (i.e. the direction where controls are effective the most; Skogestad and Postlethwaite (2005)). The higher the DCN is, the harder it is to reject the disturbance with the available controls.

The effect of wind and wave disturbance in the frequency range up to 0.3 Hz is assessed in Fig. 2, considering seven operating points of wind speed between 12 and 24 m/s. Wind turbulence acts directly on the rotor causing a variation of the aerodynamic torque, which affects rotor speed. Wind turbulence also acts on the platform, through the rotor thrust, but this excitation mechanism is less effective than wave forcing. The wind disturbance is aligned to the rotor speed output direction. Collective-pitch is very effective for controlling rotor speed, and rejecting the wave disturbance with collective-pitch does not require a large effort. This is visualized by the DCN. Wave is aligned to tower-top motion, and partially shifts towards rotor speed for increasing frequency. Waves act on the platform, but also excite the rotor response. The platform motion caused by waves produces a variation of the apparent wind speed, which affects rotor-torque, and then rotor-speed. This mechanism of excitation is more effective above the platform pitch frequency. The analysis considers zero-degree waves, that do not excite lateral motions (sway, roll, yaw). For non-zero-degree waves, also the response of these DOFs is significant and contributes to oscillations of the wind inflow. The gain of wave is maximum where wave produces the largest platform motions, so at the frequencies of platform modes, where wave excites the FOWT in resonance, and above the platform pitch frequency, where the strength of wave forcing is the maximum. The wave disturbance is not aligned to the rotor speed output direction, and the DCN shows it is very hard to counteract the wave disturbance by means of wind turbine controls.

To sum up, waves effect rotor speed, because waves drive the platform motion which result into an apparent wind speed at rotor. The wave excitation is stronger at the platform frequencies, where the FOWT is excited in resonance, and above the platform pitch frequency. Moreover, it is quite hard to counteract the wave disturbance by means of controls available in the wind turbine. Lemmer et al. (2016) carried out a similar MIMO analysis for the same floating turbine system, but based on a different linear model. They equally found that wave has a significant effect on rotor-speed and tower-top motion, and that it is difficult to counteract wave excitation by means of the control inputs available in the wind turbine. All this shows that new control strategies specific to FOWTs are needed to deal with waves. The wave-feedforward control strategy leverages the knowledge of the FOWT dynamics to improve the performance of the traditional wind turbine controller with respect to the mitigation of the wave effects.

## 5   The wave-feedforward control strategy

The wave-FF controller cancels the oscillations of rotor speed, and hence of the turbine power output, that are caused by waves. The additional collective pitch command it produces, is summed to the pitch signal of the existing generator-speed FB controller, and counteracts the variation of aerodynamic torque caused by the platform motion induced by waves. The FBFF control strategy is shown in 3.

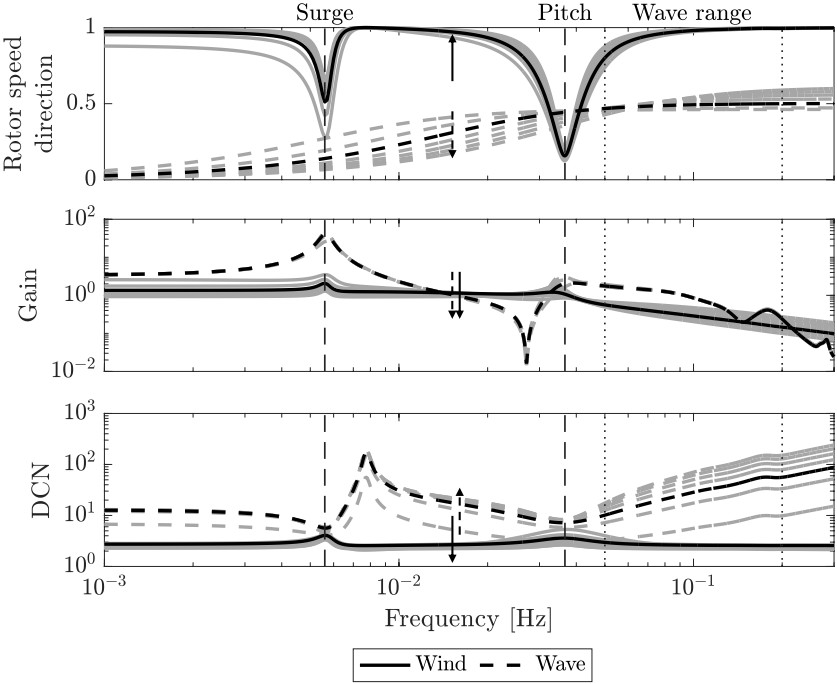

**Figure 2.** The direction with respect to the rotor speed output direction, the singular value (gain) and the disturbance condition number (DCN) associated with wind and waves. Grey lines correspond to the above-rated operating points (arrows for increasing wind) and the black line to the one of 16 m/s wind speed. The vertical dashed lines are the frequency of the platform surge and pitch modes, the frequency range where waves are active is enclosed by the vertical dotted lines.

For wave-disturbance rejection, the reference signal $r$ is zero and the closed-loop rotor speed output $\omega$ is

$$\omega = (I + G_s K_{\mathrm{fb}})^{-1}(G_s K_{\mathrm{ff}} + G_d)\eta ,\tag{12}$$

where, $G_s$ is the collective-pitch to rotor-speed plant, $G_d$ the wave disturbance model, $K_{\mathrm{fb}}$ the FB controller, $K_{\mathrm{ff}}$ the FF controller, $\eta$ the wave disturbance. In the model-inverse approach, the FF controller $K_{\mathrm{ff}}$ is designed to cancel the effect of $\eta$ on $\omega$, thus the controller transfer function is

$$K_{\mathrm{ff}} = -G_s^{-1} G_d .\tag{13}$$

$K_{\mathrm{ff}}$ is the transfer function between the input wave elevation measurement and the collective-pitch command. In general, $G_s$, $G_d$, and $K_{\mathrm{ff}}$, depend on the wind turbine operating condition and so, on the mean wind speed.

The FF controller transfer function obtained from Eq. (13) is shown in Fig. 4 for different operating conditions. There is a significant difference between the generic shape assumed by $K_{\mathrm{ff}}$ in below-rated and above-rated conditions. The amplitude is increased in below-rated winds because collective-pitch is not effective for controlling rotor speed. Here, a variation of collective-pitch produces a smaller variation of rotor torque than in above-rated winds. For this reason, it is decided to confine

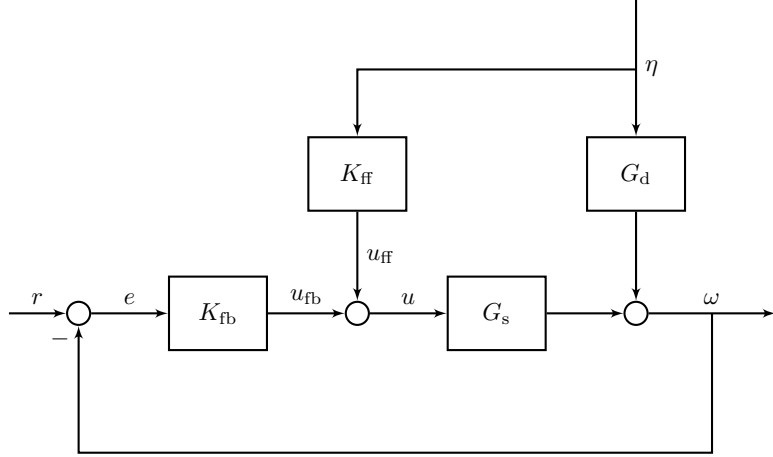

**Figure 3.** Block diagram of the feedback-feedforward controller.

the action of the FF controller to the above-rated region: when the mean pitch angle falls below a threshold, the FF action is switched-off in order to prevent excessive pitch actuators usage. The above-rated controller $K_{\text{ff}}$ has a peak at the platform pitch natural frequency which is not present in below-rated winds. In above-rated winds, the platform pitch mode damping is decreased and wave excitation leads to a large response at this frequency. This causes significant oscillations of the nacelle, with consequently large variations of the apparent wind speed, and of aerodynamic torque. A high control effort is therefore required to balance the wind fluctuations.

The control-synthesis procedure described above is valid for any platform typology. When a different platform is considered, the disturbance model changes, because forcing produced by waves depends on the platform geometry and the way waves interact with it. The FF transfer function $K_{\text{ff}}$, the product between the inverse of the plant model and the disturbance model, changes accordingly. The FF controller responds to wave, and acts in the frequency range where most of the wave energy is. In this frequency range, the amplitude of $K_{\text{ff}}$ is increased if the platform is more exposed to wave excitation, that means a larger control effort is required for wave loads rejection. The platform modes are expected to change. However, these are usually outside the wave-frequency range, and have little influence on the wave-FF action.

### 5.1 Disturbance rejection analysis

Considering the FBFF controller of Fig. 3, and the closed-loop disturbance response of Eq. (12), the FB, the FF and the FBFF sensitivity function is defined respectively as

$$S_{\text{fb}} = (1 + G_s K_{\text{fb}})^{-1} \,,$$
$$S_{\text{ff}} = 1 + G_s K_{\text{ff}} G_d^{-1} \,,$$
$$S_{\text{fbff}} = S_{\text{fb}} S_{\text{ff}} \,. \tag{14}$$

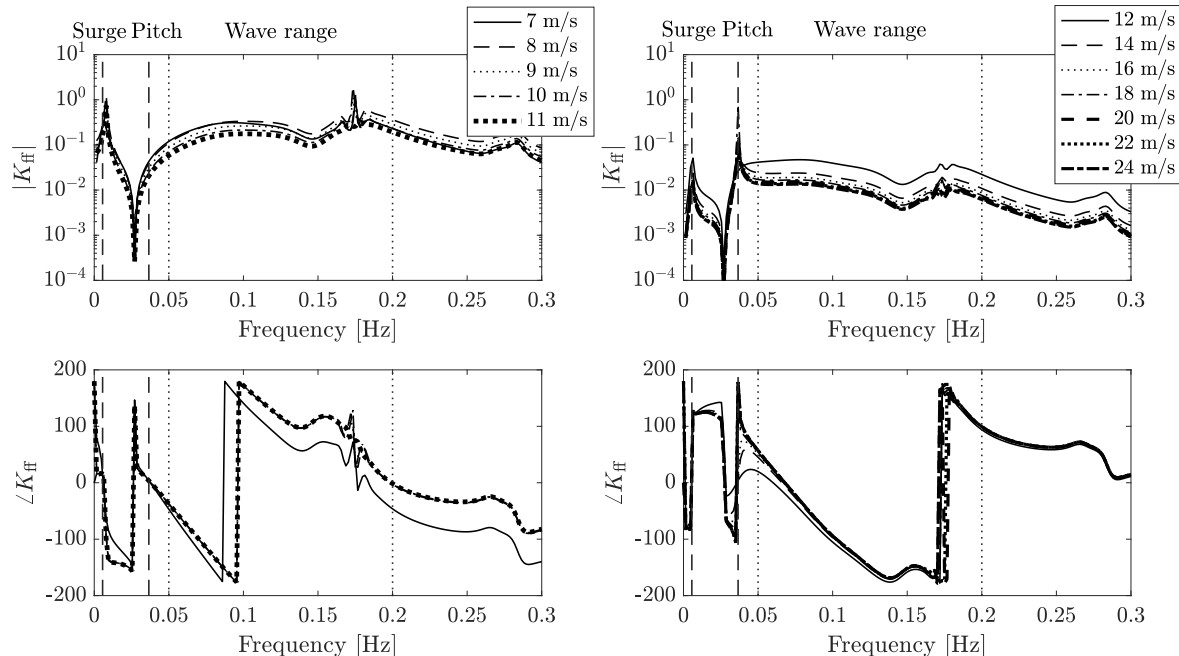

**Figure 4.** The feedforward controller transfer function $K_{\text{ff}}$ for below-rated (left) and above-rated (right) operating conditions. The vertical dashed lines are the frequency of the platform surge and pitch modes, the frequency range where waves are active is enclosed by the vertical dotted lines.

The sensitivity function of the FBFF controller is computed for different above-rated wind speeds $U_i$ to account for the different behavior of the wind turbine

$$S_{\text{fbff}}(U_i) = S_{\text{fb}}(U_i)S_{\text{ff}}(U_i) ,$$
$$S_{\text{fb}}(U_i) = (1 + G_s(U_i)K_{\text{fb}}(U_i))^{-1} ,$$
$$S_{\text{ff}}(U_i) = 1 + G_s(U_i)K_{\text{ff}}(\overline{U})G_d^{-1}(U_i) , \tag{15}$$

with $U_i = 12, 13, \ldots, 24 \ m/s$ and $\overline{U} = 16 \ m/s$. The sensitivity function tells how the disturbance is propagated to the FOWT response through the FB, FF and FBFF controllers. The lower the magnitude of the sensitivity function, the lesser the disturbance effect on the FOWT response. Figure 5 reports the sensitivity function of the FB and FBFF controllers, and compares is to the typical PSD of wind and waves (rescaled). Two curves are shown for the FB controller: one is with gains for the onshore DTU 10MW Hansen and Henriksen (2013), and one with detuned gains. In case of original gains, wind turbulence is inside the CPC bandwidth (0.074 Hz): at the controller cut-off frequency the wind spectrum is around 3% of it's maximum value. Wave loads are just above the cut-off frequency (how much above depends on the sea state). The FB controller with original gains rejects the wind disturbance, but is ineffective against wave. In case of detuned FB, the bandwidth is shorter (0.019 Hz): the wind spectrum is 18.6% of its maximum value at the controller cut-off frequency. Moreover, the disturbance sensitivity in the controller bandwidth is increased as the rotor-speed tracking performance is degraded. Hence the controller is less effec-

tive against the wind disturbance. The effectiveness of CPC with detuned gains is decreased, but detuning is needed to make the floating system stable without modifying the structure of the FB controller. As it has been shown by Larsen and Hanson (2007); van der Veen et al. (2012); Lackner (2009) when onshore tuning is utilized CPC may lead to unstable response of the

270 platform pitch mode. Bandwidth of the FB controller, and hence its effectiveness against wind turbulence, could be increased by means of NMPZ-compensation (Fischer (2013)) where pitch control is used in combination with dynamic generator-torque. Another possibility is to replace the FB controller with a more complex multivariable controller, as done by Lemmer et al. (2016). Both techniques can be used in synergy with feedforward control to further improve the floating wind turbine response to environmental loads. Interestingly, the capacity of the FB controller of rejecting wave loads is not influenced much by

275 detuning. Comparison of the sensitivity function for the FB and FBFF controller (notice that $S_{\mathrm{fbff}}$ is defined with respect to wave disturbance, whereas $S_{\mathrm{fb}}$ is valid for any disturbance), shows that sensitivity to waves is greatly reduced by addition of wave-FF.

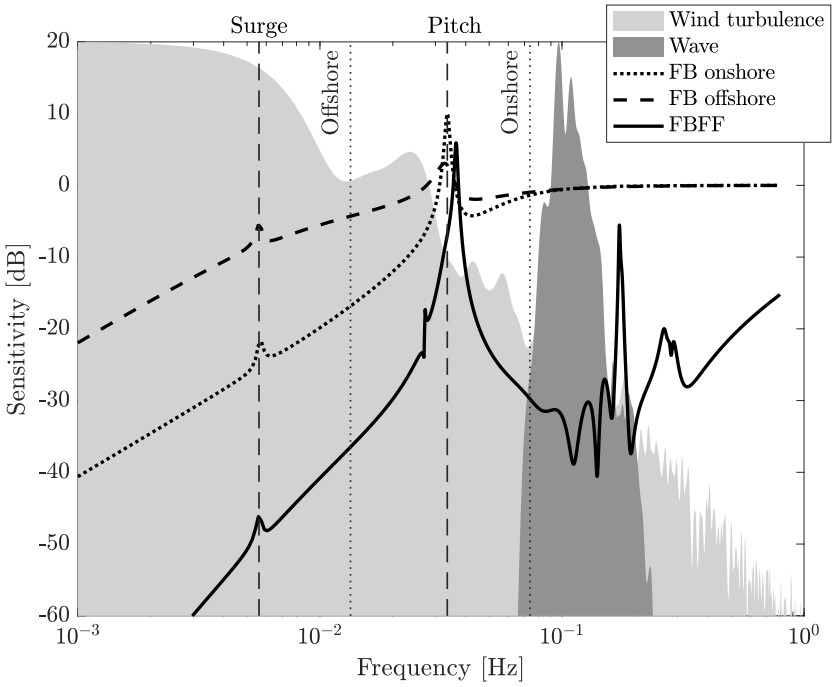

**Figure 5.** The sensitivity function of the feedback (FB) controller for onshore and offshore tuning, and of the feedback-feedforward (FBFF) in 16 m/s wind is compared to the typical PSD of wind and waves (magnitude has been rescaled to ease the comparison with sensitivity functions). The vertical dashed lines are the frequency of the platform surge and pitch modes, the dotted lines mark the bandwidth of the FB controller with onshore and offshore gains.

The disturbance-rejection function is derived from the sensitivity function and it directly relates the wave disturbance to the closed-loop rotor speed. For the FB and the FBFF controllers it is defined respectively as

$$T_{\mathrm{fb}} = S_{\mathrm{fb}} G_d \; ,$$

$$T_{\mathrm{fbff}} = S_{\mathrm{fbff}} G_d \; . \tag{16}$$

The disturbance-rejection function of the FB and FBFF controllers at 16 m/s wind speed is shown in Fig. 6 (thick solid line and thin dotted line). The magnitude of $T_{\mathrm{fb}}$ is increased in correspondence of the platform pitch mode and at higher frequencies. The disturbance-rejection function of the FBFF controller is computed for different above-rated wind speeds $U_i$

$$T_{\mathrm{fbff}}(U_i) = S_{\mathrm{fbff}}(U_i) G_d(U_i) \; . \tag{17}$$

with $U_i = 12, 13, \ldots, 24 \; m/s$ and $S_{\mathrm{fbff}}(U_i)$ already obtained in Eq. (15). The magnitude of $T_{\mathrm{fbff}}(U_i)$ is shown in Fig. 6. The disturbance-rejection function, and so the performance of the FBFF controller, is sensitive to the mean wind speed. This is due to the rotor aerodynamics which changes for different operating conditions. The benefit of wave-FF is maximum at 16 m/s, the operating point considered for model inversion, lower elsewhere, and minimum in 12 m/s wind. $T_{\mathrm{fbff}}$ is higher than $T_{\mathrm{fb}}$ around the platform pitch natural frequency. Combining the FB controller with the FF controller strengthens the coupling between platform pitch and rotor speed.

## 5.2 Gain scheduling

The FOWT dynamics (i.e. the response for a given input) depends on the mean wind speed. The turbine is more sensitive to variations of blade pitch angle in high winds, and this is visualized in the input-output analysis of Fig. 1. As shown in Fig. 2, sensitivity to waves remains constant, and does not depend significantly on wind speed. The control effort required to counteract a given wave is different depending on wind speed, because the wind turbine responds in a different way to blade-pitch angle variation. Intuitively, the pitch action required to reject the effects of a given wave is lower in high winds. To have the maximum possible reduction of the wave disturbance, the FF controller needs to consider how the FOWT dynamics are modified with operating condition, and a gain-scheduling strategy is introduced for this purpose.

Based on the procedure introduced above, a linear model of the FOWT is computed for several above-rated wind speeds and, by means of Eq. (13), an FF controller is obtained for each of them. The transfer function of the FF controllers is shown in Fig. 7. From visual inspection of the figure, it is evident the effort required to cancel the wave disturbance is maximum in near-rated winds and decreases in high winds. If the FF controller obtained from the 16 m/s model is used at any wind speed, the FF action would be less-than-ideal for wind speeds between rated and 16 m/s, and higher-than-ideal for greater wind speeds, leading to a decreased performance, as highlighted by the disturbance rejection analysis of Fig. 6.

Figure 7 also reveals the shape of the FF controller does not change much with wind speed except for the static gain. Based on this consideration, the performance of the FF controller is improved by adjusting the static gain based on the actual turbine operating condition. In other words, a single FF controller is computed for the 16 m/s condition and the static gain is modified as wind speed changes, to reflect the changed dynamics of the FOWT. The actual collective-pitch angle is chosen as the

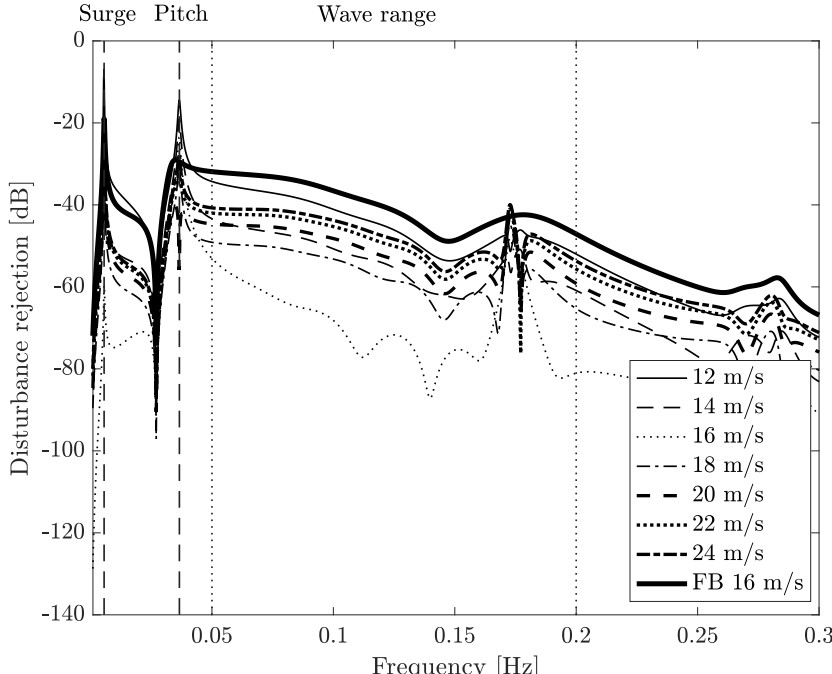

**Figure 6.** Disturbance-rejection function of the feedback-feedforward controller in above-rated winds. The feedback controller (FB) in 16 m/s wind is reported for comparison. The vertical dashed lines are the frequency of the platform surge and pitch modes, whereas the frequency range where waves are active is enclosed by the vertical dotted lines.

scheduling variable. The gain-scheduling law is obtained fitting a quadratic function to the DC-gain of the $K_{ff}(j\omega)$ computed

for different above-rated winds. The scheduled FF controller is

$$K_{ff}(\beta) = c_{ff}(\beta)K_{ff}(\overline{U}) \, ,$$
$$c_{ff}(\beta) = p_2\beta^2 + p_1\beta + p_0 \, , \tag{18}$$

where $p_2, p_1, p_0$ are the coefficients of the quadratic best-fit function, $\beta$ is collective-pitch, and $K_{ff}(\overline{U})$ is the FF controller for 16 m/s wind speed.

In Fig. 7, the scheduled FF controllers $K_{ff}(\beta) = c_{ff}(\beta)K_{ff}(\overline{U})$ are compared to the model-inversion FF controllers obtained

from the evaluation of Eq. (13) for different above-rated wind speeds. The scheduled controller is a good approximation of the ideal case. The proposed scheduling strategy leaves the phase of $K_{ff}(j\omega)$ unchanged, but this is acceptable since the phase does not change much with wind speed.

The disturbance rejection function of the FBFF controller with scheduling is obtained by replacing $K_{ff}(\overline{U})$ with $K_{ff}(\beta)$ in Eq. (15) and Eq. (17) and it is shown in Fig. 8. The disturbance rejection in the wave frequency range is lower for any wind

speed, as the controller action is adjusted based on the wind turbine operating condition.

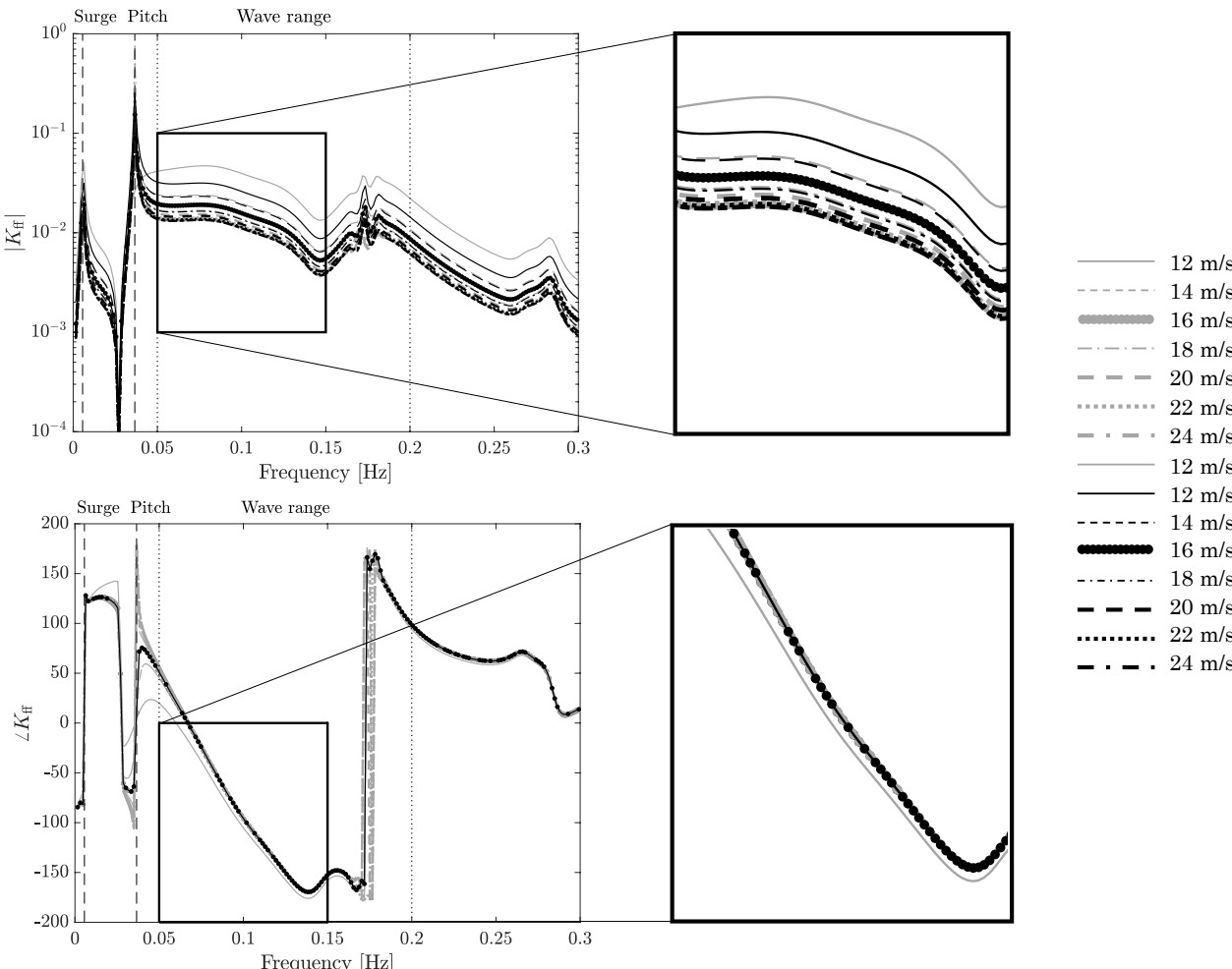

**Figure 7.** The scheduled feedforward controllers $K_{\mathrm{ff}}(\beta) = c_{\mathrm{ff}}(\beta)K_{\mathrm{ff}}(\overline{U})$ (black) obtained from the scheduling of the 16 m/s controller $K_{\mathrm{ff}}(\overline{U})$ (dotted line) are compared to the model inversion controllers $K_{\mathrm{ff}}(U_i)$ (grey) for different above-rated wind speeds $U_i = 12, 13, \ldots, 24\ m/s$. Magnitude (top) and phase (bottom). Wave range is the frequency range where linear wave is active.

The FF controller for implementation is obtained as in Eq. (18). The order of the transfer function $K_{\mathrm{ff}}(\overline{U})$ is too high for practical usage: a reduced-order approximation is utilized in place of the original transfer function. The low-pass filtered collective-pitch angle measurement is used for scheduling.

## 6 Wave measurement and prediction

The transfer function of the FF controller has an intrinsic delay of $t_d$. A suitable wave elevation measurement is required to compensate the intrinsic delay of the FF controller: it is required to know the wave $t_d$ before it arrives at the platform. The

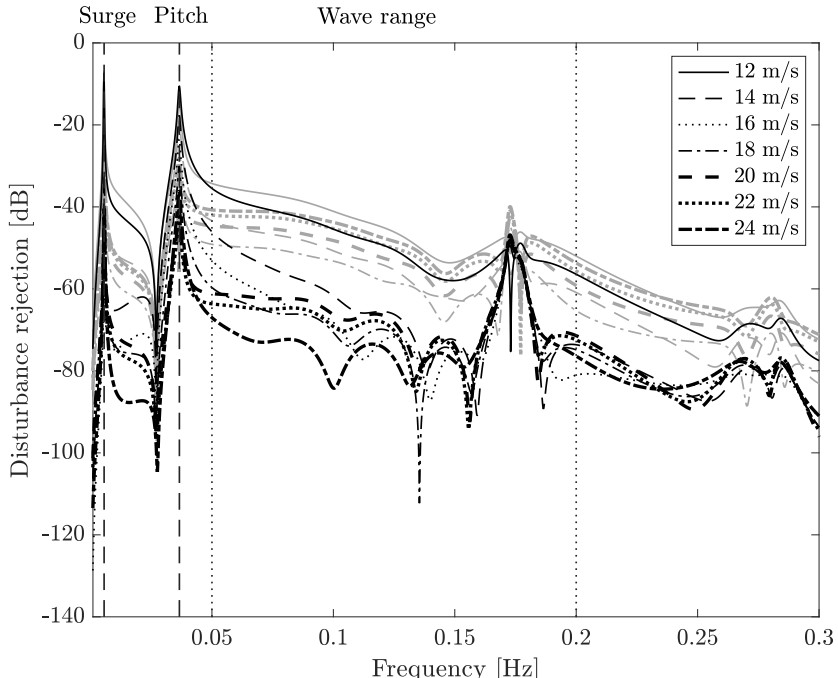

**Figure 8.** Disturbance rejection of the FBFF controller with scheduling (black) and without (grey), for several above-rated operating conditions. The vertical dashed lines are the frequency of the platform surge and pitch modes, the frequency range where waves are active is enclosed by the vertical dotted lines.

wave prediction is obtained from a measurement of the surface elevation in a point at a distance $l$ upstream the platform. The measurement is propagated downstream in space and forward in time.

The wave elevation in two points along the wave propagation direction is related by the frequency response function

$$H_l = e^{-jkl}, \tag{19}$$

where $k$ is the wave number, and, for gravity waves on deep-water, $k \approx \omega^2/g$.

The frequency response function that relates the upstream wave measurement to the wave prediction at the FOWT is

$$H(j\omega) = e^{j\omega\left(t_d - \frac{\omega l}{g}\right)}. \tag{20}$$

For a given distance $l$ and a preview time of $t_d$, $H(j\omega)$ behaves as a pure negative-delay operator only for $\omega > gt_d/l = \omega_t$. The wave spectral components with a frequency greater than $\omega_t$ are successfully predicted. Prediction of the wave components with a lower frequency is not possible, because the wave arrives at the platform location in a time lower than the preview time $t_d$. It is possible to predict the lower-frequency harmonics by measuring the wave elevation far upstream the FOWT or by decreasing the preview time.

For real-time control purposes, the wave prediction model of Eq. (20) is implemented as it is shown in Fig. 9. The wave

elevation $\eta$ is continuously measured in $l$ meters upstream the floating wind turbine with a sample rate $T_s$. At any time instant $t_k$, the discrete Fourier transform (DFT) of the last $n$ samples is computed to obtain the complex spectrum $y$. The element-wise product between $y$ and the transfer function $H$, evaluated at $n$ discrete frequencies $\omega_i = (2\pi i/nT_s)$, $i = 0, \ldots, n-1$, gives the spectrum of the predicted wave elevation at platform location $\hat{y}$. The Inverse DFT of $\hat{y}$ gives $\hat{\eta}$, which is the wave elevation that is going to be experienced by floating wind turbine $d = t_d/T_s$ time samples ahead in time.

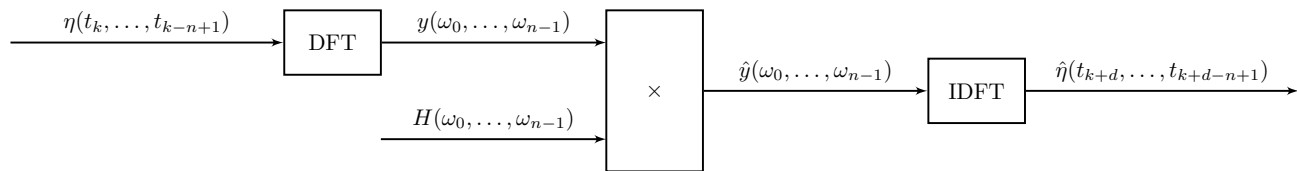

**Figure 9.** Scheme of the wave prediction algorithm. $\eta$ is the wave elevation measurement upstream the floating wind turbine and $\hat{\eta}$ is the wave elevation at platform location, $d = t_d/T_s$ time samples ahead in time.

Several technologies are available to measure the surface elevation. Some examples are wave-rider buoy, radar, airborne or satellite. The radar technology is particularly attractive because it scans a large area, it detects waves far from its location (up to 4 km) and it is capable of fully autonomous operation. The X-band radar, commonly used by ships for navigation, received a lot of attention as a remote wave sensor. Images of the wave field are obtained from the radar as radar beams are reflected and shadowed by the crests of the wave fronts. An example of this technology is the wave monitoring system WaMoS II introduced

by Ziemer and Dittmer (1994) and at the base of the real-time wave-prediction system developed by Reichert et al. (2010) within the On board Wave and Motion Estimator (OWME). A methodology based on 2D-FFT is proposed by Naaijen and Wijaya (2014) to obtain a directional phase-resolved prediction of the wave elevation from radar data (additional information about the directional energy spectrum is required, e.g. from a wave buoy). A similar measurement could be used in wave-FF control.

**7    Results**

The wave-FF control strategy is evaluated by means of numerical simulations in the servo-aero-hydro-elastic code FAST (Jonkman and Marshall (2005)). The FAST model has 7 DOFs: platform motions (surge, sway, heave, roll, pitch and yaw) and the rotor rotation. The drivetrain is rigid as well as the tower and blades. The hydrodynamic model is based on linear potential flow theory with viscous effects. The radiation and the first-order wave forces are computed prior to the simulation based on

the same WAMIT data that are used to build the control-design model. The calculation of the frequency-dependent radiation loads is based on the convolution integral of the retardation functions matrix. Second-order wave loads are modeled by means of the approximation technique introduced by Newman (1974).

The wave-FF control strategy considered for the verification is displayed in Fig. 10. Three cases are considered: a baseline case with only FB control, the FBFF control without gain scheduling, and the FBFF control with gain scheduling. In the simulations, an ideal upstream wave measurement is used and the accuracy of the wave measurement system (e.g. radar) is not taken into account. Results are therefore indicative of the upper performance limit of wave-FF control.

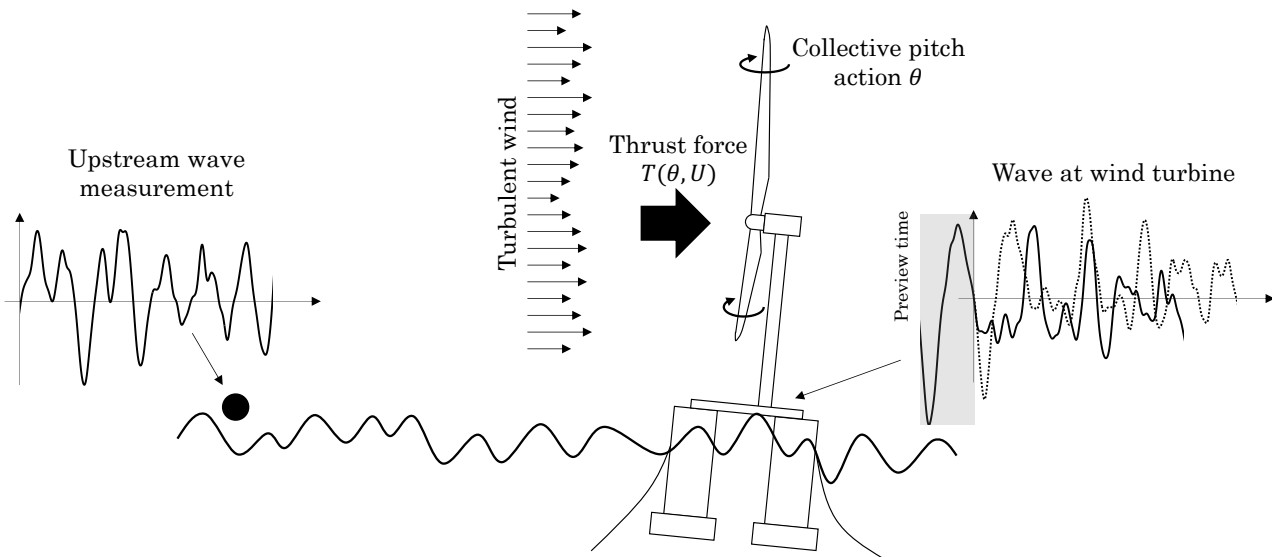

**Figure 10.** Schematics of the wave-feedforward control strategy. Wave excite the floating platform and generate a varying apparent wind speed for the rotor. The oscillating wind results into rotor speed fluctuations which are only partially rejected by the standard feedback controller. The feedforward action is based on the wave elevation measured upstream the wind turbine. This measurement is used to obtain a preview of the wave elevation at the floating platform, which is the input of the controller. The resulting collective-pitch action, which is summed to the pitch request from the feedback controller, counteracts the wave disturbance, modifying the aerodynamic torque and the rotor thrust force.

## 7.1 Environmental conditions

Three realistic turbulent wind and irregular wave combinations (see table 1), representative of an offshore site with moderate-severity met-ocean conditions, are selected for assessing the performance gains of the wave-FF control strategy. The reference offshore site is part of the Gulf of Maine (North Atlantic ocean), about 25 km southwest of Monhegan Island 65 km east of Portland, and the mean water depth is 130 m. Met-ocean data for the site are reported by Gonzalez et al. (2015). Three above-rated winds are considered, that are defined by the parameters of Tab. 1. For any condition, wave was defined according with the Pierson-Moskowitz spectrum. The significant-heigh and peak-period were selected as the most probable combination of values for the assigned wind speed. Wind and waves are aligned to the zero-degree direction.

**Table 1.** Met-ocean conditions considered for the verification of the wave-FF control strategy.

| Hub-height mean wind speed [m/s] | Turbulence intensity [%] | Significant wave height [m] | Wave peak period [s] |
|---|---|---|---|
| 16 | 12 | 1.5 | 10.0 |
| 18 | 11 | 2.5 | 10.0 |
| 22 | 10 | 3.5 | 8.0 |

## 7.2 Wave prediction

The wave prediction algorithm presented in section 6 is tested in the met-ocean conditions corresponding to the 16 m/s mean wind speed case of table 1. The wave elevation is sampled every 0.1 s at distance of 200 m upstream the FOWT. The wave elevation at platform location is computed based on the last 1000 samples and a preview time of 7.5 s is requested. The wave elevation preview is compared to the wave at platform in Fig. 11. The overall quality of the estimate is good. The PSD of the two signals reveals that the largest error is introduced in the low-frequency harmonics. The error is due to the intrinsic characteristics of the transfer function on which the wave prediction algorithm is based. For the present case, the transfer functions correctly predicts the wave harmonics above a threshold frequency of 0.058 Hz.

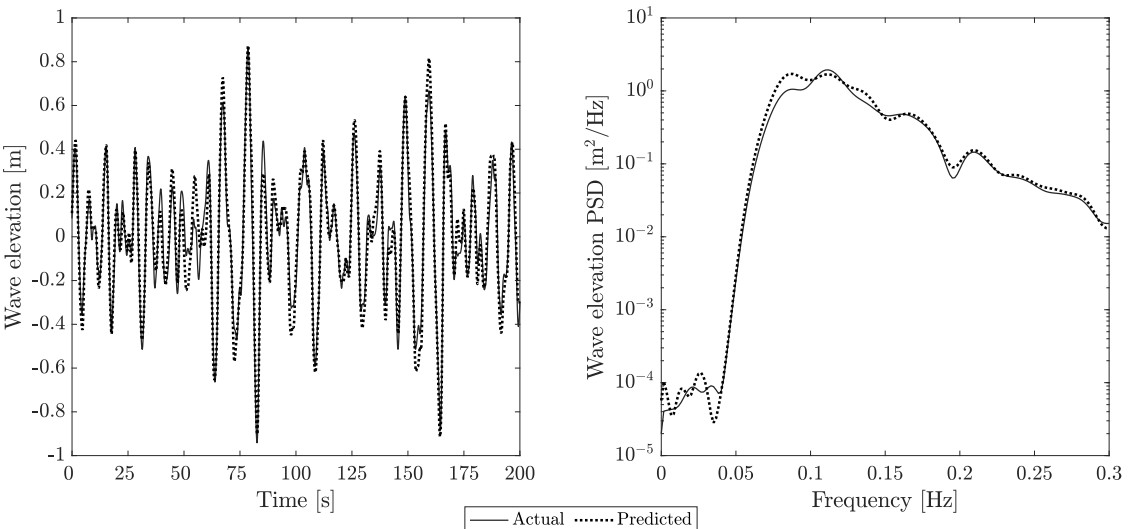

**Figure 11.** Left: the wave preview obtained by means of the prediction algorithm is compared with the wave at platform location (in the plot, the preview is delayed of the preview time $t_d = 7.5s$ to ease the comparison). Right: the wave preview and measurement are compared based on their PSD.

### 7.3 Steady wind

The effect of the FF control strategy is first demonstrated considering a steady wind without shear. With this assumption, the
385 wave is the only disturbance acting on the FOWT. Sample time series of the rotor speed and blade pitch command for a 22 m/s wind speed case are shown in Fig. 12. The amplitude of rotor speed oscillations caused by the wave disturbance is reduced with FBFF control with respect to the FB case, and this is achieved at the expense of an increased pitch activity. The pitch effort required by the scheduled FBFF is less than without scheduling for a comparable disturbance rejection performance.

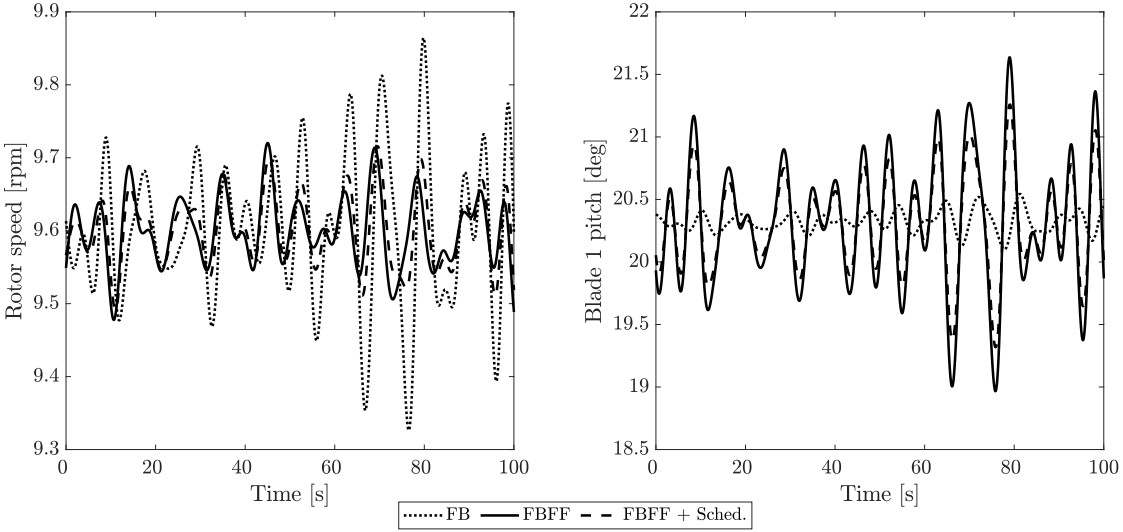

**Figure 12.** Time series of the rotor speed (left) and blade 1 pitch angle command (right) for the 22 m/s steady wind case.

### 7.4 Turbulent wind

The FBFF control is evaluated in more realistic power production conditions. Turbulent wind fields were generated in `turbsim` (Kelley and Jonkman (2007)) with Kaimal spectrum, a power-law profile with exponent 0.14, and turbulence intensity selected for each mean wind speed according with the IEC-61400 (class IC turbine). For every condition of Tab. 1, six independent wind-wave realizations, each 10-minutes long, were considered, as recommended by the IEC 61400 standard (IEC (2005)) to get statistically-significant data. An initial pre-simulation time of 1000s was included at the beginning of each simulation and
cut-out from results to exclude initial transients.

Sample time series of rotor speed and blade-pitch angle for the 22 m/s case are shown in Fig. 13. As visible looking at the FB case, the largest fraction of rotor speed oscillations is due to wind turbulence. This is in agreement with the MIMO disturbance analysis of section 4. The FF control reduces the part of rotor speed oscillations caused by waves, but it does not compensate for the effect of wind turbulence. The pitch actuation is increased with any FBFF compared to the FB case.

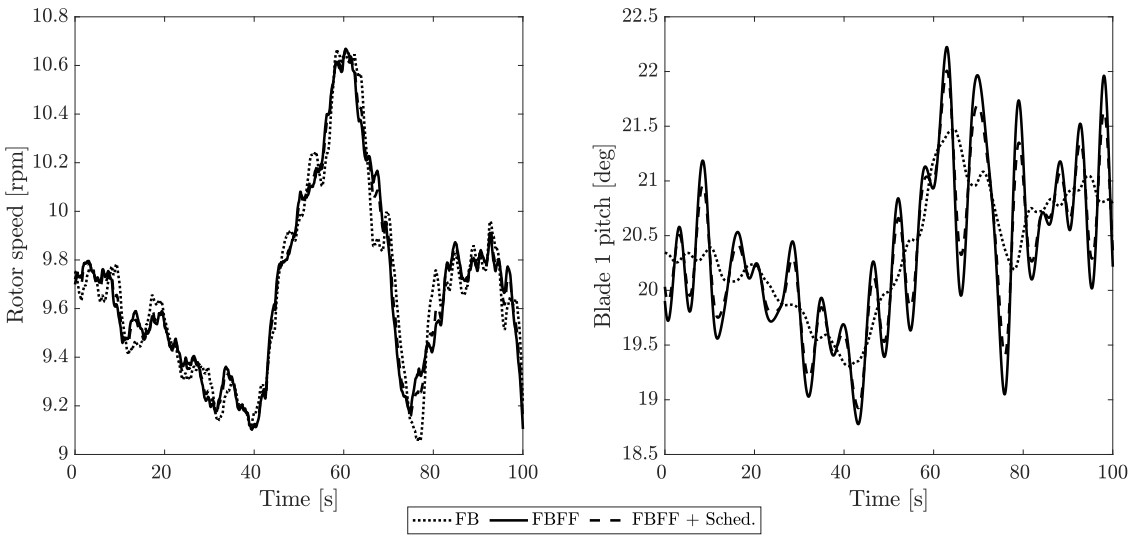

**Figure 13.** Time series of rotor speed (left) and blade 1 pitch-angle-command (right) for the 22 m/s turbulent wind.

Power spectral density (PSD) of rotor speed and pitch-angle-command is computed from the aggregated time series of the six seeds relative to the same operating condition. Results for the 22 m/s case are reported in Fig. 14. FBFF clearly reduces the energy content of rotor speed associated with wave excitation (wave range, 0.05-0.2 Hz), but has a negligible effect at lower frequency, where rotor speed oscillations are in large part due to wind turbulence. The FF action results into an additional blade-pitch command which energy content is concentrated in the wave range. Rejection of wave disturbance is slightly improved by the addition of scheduling (FBFF + Sched.), which requires also a lower blade-pitch effort compared with FBFF.

Wave-FF is designed to reduce the effects of wave disturbance and improve rotor speed regulation. However, it is expected to affect also the structural loads for the turbine components. Fatigue loads for each operating condition are evaluated in terms of damage-equivalent loads (DEL) computed with `mLife` (Hayman and Buhl (2012)). Wave-FF is also expected to affect platform motions, and this is quantified by the standard deviation of platform surge, roll and pitch. DEL and standard deviations are computed for every operating condition based on the aggregated time series of six seeds. Variation of DEL and standard deviations with FBFF + Sched. compared with FB is examined in Fig. 15 for the three load cases. The LSS-torque DEL is reduced up to 16%. The LSS-torque depends on the aerodynamic and generator torque. The wave-FF reduces the aerodynamic torque oscillations caused by waves and so the fatigue loads for the LSS. The lower dynamics of LSS-torque is reflected in the platform roll-motion which is reduced as well. The highest LSS-torque DEL reduction is achieved in high winds when waves are the strongest. Blade pitch is increased because of the additional wave-FF command, and the increment is proportional to the strength of waves. As demonstrated in the input-output analysis of §4, aerodynamic torque and rotor thrust are both affected by blade-pitching. The modification of the thrust force induced by wave-FF impacts the along-wind platform motions and the tower-FA loads. Platform surge and pitch motions are increased, with consequently higher fatigue loads for the mooring system, and lower loads for the tower. It is counterintuitive, but larger motions imply lower tower loads.

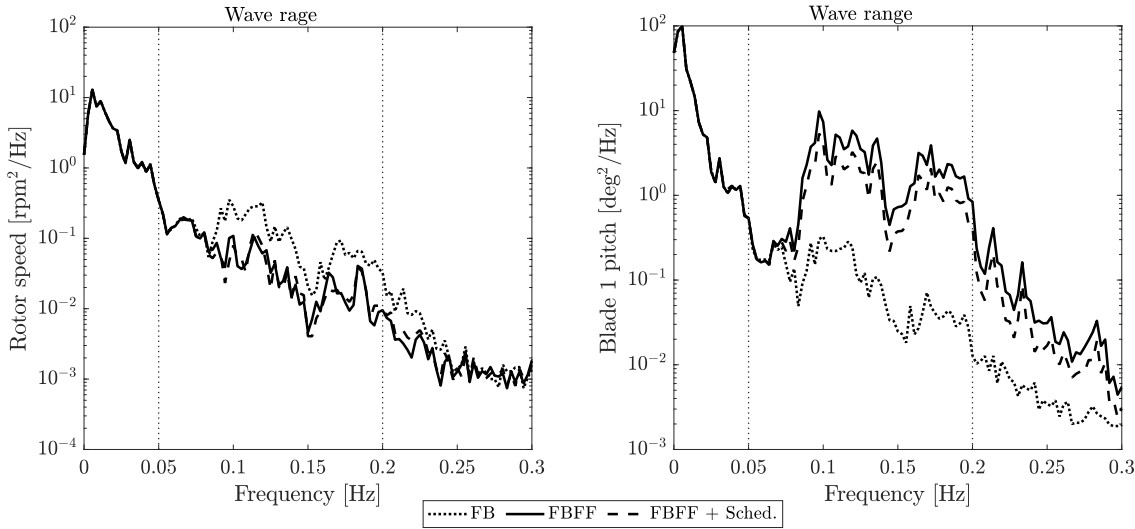

**Figure 14.** Power spectral density of rotor speed (left) and blade 1 pitch angle command (right) aggregated time series of six seeds for the 22 m/s turbulent wind case. Wave range is the frequency range where linear wave is active.

As noticed by Fleming et al. (2016), a smooth motion of the nacelle releases the energy introduced by waves, which therefore is not put into tower bending.

## 8 Conclusions

This paper investigated a model-inversion feedforward control strategy for mitigation of wave excitation in floating offshore wind turbines. A linear control-design model is utilized to carry out an MIMO analysis of the floating wind turbine. Collective-pitch is more effective than generator torque for controlling rotor-speed in above-rated winds. Above the platform natural frequencies, wave equally affects rotor and platform motions, with the same strength of wind turbulence. Based on linear analysis, a model-inversion feedforward controller is designed for canceling the wave-induced rotor-speed (and generator-power) oscillations using collective-pitch. The feedforward controller is added to an industry-standard feedback controller and the performance improvement is demonstrated by means of linear analysis. A gain-scheduling algorithm is devised to improve the controller performance by adapting the feedforward action as the wind turbine operating condition changes. The control strategy is finally verified by means of time-domain simulations in a non-linear aero-servo-hydro-elastic model. It is found that feedforward control can reduce the standard deviation of rotor speed up to 2%. It also has a positive side effect on the fatigue loads of several wind turbine components: the shaft torsion is reduced up to 16%, the tower-base fore-aft bending up to 5%. Platform motions are slightly increased, and this is reflected into the mooring line loads. The blade-pitch actuator usage is increased. Wave-FF control improves the dynamic response of the floating turbine without requiring the replacement of the industry-standard feedback controller. A wave-measurement and forecast system must be implemented, but this is feasible to

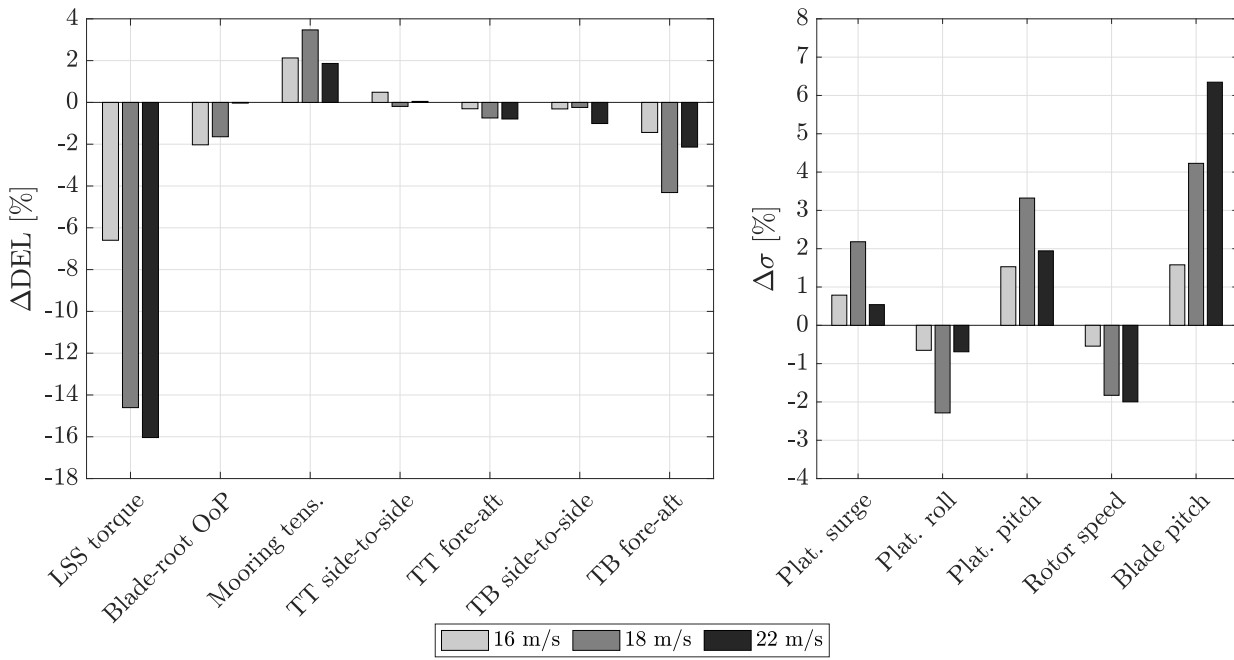

**Figure 15.** Damage equivalent load (DEL) and standard deviation ($\sigma$) with FBFF + Sched. for three above-rated power production conditions. Statistics for every operating conditions were obtained from aggregated time series of six seeds. A negative $\Delta$ means a reduction with respect to FB-only control. LSS stands for low-speed shaft, OoP for out-of-plane, TT for tower-top and TB for tower-base.

get with technologies already used in the maritime industry. The extra cost of the wave-measurement system, and fatigue of the blade-pitch actuators is likely to be offset by the lower cost of the turbine generator and tower, which can be redesigned in reason of the reduced overspeed and fatigue loads, respectively.

The following suggestions should be considered in future work about wave-based and wave-feedforward control in floating wind turbines:

– the wave-feedforward controller is sensitive to accuracy of wave-elevation prediction and to fidelity of the wave distur-bance model. The focus of this work is about development of the wave-feedforward control strategy and did not addressed the topic of uncertainties In the present work, uncertainties in the wave measurement are only due to the preview algo-
rithm and, as it is shown, the prediction error is small. Larger errors are expected when using a realistic measurement of upstream waves. Model uncertainties are mostly related to identification of the wave-excitation model.The model we consider here has been assessed against a medium-fidelity model in a previous work (Fontanella et al. (2020)) and was deemed sufficiently accurate for control-design. In-depth analysis of the controller robustness with respect to model and measurement uncertainties should be addressed in future works. Implementing the wave-FF control in FOWTs based on
different platform typologies may reduce uncertainty about the benefits of this control strategy;

- the proposed feedforward controller is linear and compensates only for first-order wave loads. Recent numerical and experimental studies, for example the one of Roald et al. (2013), prove that second-order wave loads have a noticeable effect on a floating turbine response. Thus, a possible research suggestion is to investigate non-linear controllers and to include second-order hydrodynamics in the controller design;

- in the control-design model, rotor aerodynamics are modeled based on the quasi-steady theory. Thus, the controller obtained from the model does not account for unsteady aerodynamic effects, which may be significant for the response in the upper wave-frequency range (Mancini et al. (2020)). It is therefore suggested to develop a control-oriented model of the unsteady rotor aerodynamics and to include it in the control-design model, so to investigate how unsteadiness affects the response of the controlled FOWT;

- in the case at hand, the feedforward controller is designed to regulate rotor speed, and the reduction of tower loads is obtained as a positive side effect. A large fraction of tower loads is caused by wave, so it is advisable to use wave information to reduce tower fatigue loads;

- the wave prediction model may find application in several control-related tasks, which are not envisioned here. Waves drive the rigid-body motion of the floating turbine, and this is likely to affect the turbine wake (Wise and Bachynski
(2020)). Wave prediction may be included in future floating wind farm control strategies;

- single-input single-output feedback controllers remain the default choice in floating wind turbines and advanced controllers are still far from reaching commercial projects. Tighter relationships between industry and academia are advisable to promote the adoption of advanced control strategies.

*Author contributions.* AF and MA developed the wave feedforward control methodology in all its aspects. MB and JWvW supervised the
470 research activity mentoring AF and MA. Finally, AF prepared the manuscript of this article with contribution from all co-authors.

*Competing interests.* The authors declare that they have no conflict of interest.

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
