# Peer review of "Model-based design of a wave-feedforward control strategy in floating wind turbines"

_Wind Energy Science, 2021_

## Author Comment (AC1)

[Figure]

Figure 1: The sensitivity function of the blade-pitch feedback controller with original (Onshore) and detuned (Offshore) gains. In the background, it is shown the typical PSD of wind and waves, which magnitude has been rescaled to ease the comparison with sensitivity functions.

---

## Author Response (AR1)

Politecnico di Milano
Department of Mechanical Engineering
Via La Masa 1, 20156, Milan
Italy

Wind Energy Science Discussion

Date: April 29, 2021
Subject: WES-2021-9 Final Response

Dear Referees,

We would like to express you our gratitude for taking the time to review our manuscript and for the careful feedback you have provided. We truly think your comments have helped us improving our work, that has been enriched and is now clearer and more attractive for the wind energy community.

In the rest of this document, you can find the detailed response to your comments, and the new version of the article that has been revised according to your suggestions.

On behalf of all Authors,
yours sincerely,

Alessandro Fontanella

Attached documents:
- Response to Anonymous Referee #1
- Response to Farid Khazaeli Moghamad (Referee #2)
- Manuscript changes

**Response to Anonymous Referee #1**

Dear Referee, thank you very much for your comments and feedback.

Much of the work about floating wind turbine control carried out so far focus on the interaction between blade pitch controller and platform pitch motion, and how to avoid the negative-damping issue. The Authors are grateful to Referee #1 for introducing this topic in the discussion and believe that clarifying how the pitch controller tuning relates with wave disturbance rejection makes the research more interesting for both the scientific community and industry.

| | |
|---|---|
| **RC1.1** | Line 63: Does this tuning [of the feedback pitch controller] imply that the bandwidth of the blade pitch controller is reduced below the natural frequency of the floater pitch motion? (Arriving at 0.0186Hz) If yes, could you elaborate on the effect of this tuning which seems to give a controller bandwidth that is reduced with a factor ~4-5 compared to a conventional controller for bottom fixed wind turbines. With respect to aerodynamic loads within the bandwidth of a conventional controller but not within the bandwidth of the tuned controller, and with respect to the effectiveness of the proposed wave disturbance rejection. |
| **AC1.1** | We appreciate the Referee's comment because it helps clarifying why the industry-standard speed-regulation control strategy is not effective in rejecting the wave disturbance. In case of original gains, wind loads are inside the collective-pitch controller (CPC) bandwidth: at the controller cut-off frequency the wind spectrum is around 3% of its maximum value. Wave loads are just above the cut-off frequency (how much above depends on the sea state). The CPC with original gains rejects the wind disturbance but is ineffective against wave. In case of detuned gains, the bandwidth is shorter: the wind spectrum is 18.6% of its maximum value at the controller cut-off frequency. Moreover, the disturbance sensitivity in the controller bandwidth is increased, as rotor-speed tracking performance is degraded. Hence the controller is less effective against the wind disturbance. The capability or rejecting wave loads is not influenced much by detuning. |
| | This is exemplified by the figure below, where the typical PSD of wind and waves (rescaled) is compared to the sensitivity of the feedback pitch controller with original and detuned gains. The feedforward controller complements the feedback CPC, and targets wave loads. Therefore, its benefits are weekly related to the tuning of the feedback controller. |

[Figure]

*Figure 1. The sensitivity function of the feedback (FB) controller for onshore and offshore tuning, and of the feedback-feedforward (FBFF) in 16 m/s wind is compared to the typical PSD of wind and waves (magnitude has been rescaled to ease the comparison with sensitivity functions). The vertical dashed lines are the frequency of the platform surge and pitch modes, the dotted lines mark the bandwidth of the FB controller with onshore and offshore gains.*

The reasoning above has been included in the section "The wave-feedforward control strategy – Disturbance rejection analysis".

**RC1.2**  Line 84: Something seems to be wrong with the reference.

**AC1.2**  We agree with Referee that reduced-order modelling of an FOWT rotor is not addressed sufficiently well by the paper cited in the manuscript ("Robust gain scheduling baseline controller for floating offshore turbines", F. Lemmer, 2020). A more appropriate reference about the topic is "Multibody modelling for concept-level floating offshore wind turbine design" (F. Lemmer, 2020).

**RC1.3**  Line 390: "As it has been shown, a large part of rotor speed oscillations is caused by wind turbulence" Could some of these variations and associated DELs have been reduced if a conventional (bottom fixed) controller bandwidth had been applied?

**AC1.3**  We appreciate this comment because allows us to discuss the synergy between the feedforward control strategy we propose, and the feedback collective-pitch controller (CPC) currently used in commercial wind turbines. The effectiveness of CPC with detuned gains is decreased, but detuning is needed to make the floating system stable without modifying the structure of the FB controller. The bandwidth of the FB controller, and hence its effectiveness against wind turbulence, could be increased by means of NMPZ-compensation (B. Fischer, 2013), where pitch control is used in combination with dynamic generator-torque. Another possibility is to replace the FB controller with a more complex multivariable controller, as done in "Control design methods for floating wind turbines for optimal disturbance rejection" (F. Lemmer, 2016). Both these techniques can be used in synergy with feedforward control to further improve the floating wind turbine response to environmental loads.

**Response to Farid Khazaeli Moghamad (Referee #2)**

Dear Farid, thank you very much for your accurate comments. We tried to answer all of them at our best, because we are sure it will improve the quality of the article. From the level of detail of the comments we are sure you spent a good amount of time in critically reviewing our manuscript and we would like to thank you for that.

**RC2.1**    2- The proposed feedforward approach seems to be very sensitive to the wave elevation prediction and wave disturbance model. Have you checked the influence of uncertainties in the wave elevation prediction and wave disturbance model on the proposed controller performance?

**AC2.1**    Thank you, Farid, for this comment: sensitivity to disturbance measurement and model uncertainties is very important in feedforward control in general. As you correctly pointed out, the wave-feedforward controller is sensitive to accuracy of wave-elevation prediction, and to fidelity of the wave disturbance model. The focus of this work is about development of the wave-feedforward control strategy and did not addressed the topic of uncertainties. In the present work, uncertainties in the wave measurement are only due to the preview algorithm and, as it is shown, the prediction error is small. Larger errors are expected when using a realistic measurement of upstream waves. Model uncertainties are mostly related to identification of the wave-excitation model. The linear-model we consider here has been assessed against a medium-fidelity model in a previous work and was deemed sufficiently accurate for control-design. In-depth analysis of the controller robustness with respect to model and measurement uncertainties should be addressed in future works.

**RC2.2**    3- The methodology employed for damage equivalent load estimation in Section 7.4 needs to be explained in more detail. How are the equivalent loads and stresses estimated?

**AC2.2**    Fatigue loads were evaluated in terms of damage-equivalent loads (DEL) computed with the standard tool `mLife` based on aggregated 10-minutes time series of six different seeds. We added this clarification to the article. The authors refer readers interested in the algorithm used by `mLife` to the software guide, rather than including (partial) information about it in the article.

**RC2.3**    4- Please provide more details about the simulation studies in Section 7.4. How long is the time duration of the data set associated to the results in fig. 14? Is it enough to capture the wave and wind dynamics? Are the estimated DEL and standard deviation only based on a single data block of each operating speed?

**AC2.3**    Thank you for this comment: information about the simulation setup were needed to understand results about loads and motions. As the Referee pointed out, the wind turbine loads may depend significantly on the simulation time. For every operating condition considered in the analysis (Tab. 1), six independent wind-wave realizations, each 10-minutes long, were considered, following the recommendation of the IEC 61400 standard. An initial pre-simulation time of 1000s was included at the beginning of simulations and cut-out from results so to exclude initial transients. Power spectral densities were computed from the aggregated time series of the six seeds relative to the same

operating condition. Also, damage-equivalent loads and standard deviations were computed for every operating condition based on the aggregated time series of six seeds. This information has been reported in the article section "Results – Turbulent wind". Moreover, PSDs reported in the manuscript were based on a single wind/wave seed (10-min data). New PSDs were computed based on six seed and the result is statistically more significant.

| | |
|---|---|
| **RC2.4** | 5- Why are the platform motions increased in case of using FBFF with scheduling compared to FB? have you also been looking into the platform rolling? |
| **AC2.4** | Thank you for this comment. We appreciated it because it allows us to discuss better the working principle of the feedforward controller and, maybe more importantly, the coupled rotor-platform dynamics that is very significant for the response of any floating turbine. The low-speed shaft (LSS) torque depends on the aerodynamic and generator torque. The wave-feedforward reduces the aerodynamic torque oscillations caused by waves and so the fatigue loads for the LSS. The lower dynamics of LSS-torque is reflected in the platform roll-motion which is reduced as well. As demonstrated in the input-output analysis of section 4 "FOWT response to controls, wind and waves", aerodynamic torque and rotor thrust are both effected by collective pitch. The modification of the thrust force induced by wave-FF impacts the along-wind platform motions, surge and pitch, that are increased with FBFF compared to FB. |
| **RC2.5** | 6- The gain scheduling algorithm is tuned based on different intervals of wind speed as one of the system exogenous inputs, but what about wave elevation as the other disturbance input? |
| **AC2.5** | The FOWT dynamics (the response for a given input) depends on the mean wind speed. The turbine is more sensitive to variations of blade pitch angle in high winds, and this is visualized in the input-output of section 4. As shown in Fig. 1, sensitivity to waves remains constant, and does not depend significantly on wind speed. The control effort required to counteract a given wave is different depending on wind speed, because the wind turbine responds in a different way to blade-pitch angle variation. Intuitively, the pitch action required to reject the effects of a given wave is lower in high winds. To have the maximum possible reduction of the wave disturbance, the FF controller needs to consider how the FOWT dynamics are modified with operating condition, and a gain-scheduling strategy is introduced for this purpose. The FOWT input-output dynamics does not depend on the magnitude of the wave elevation, thus the scheduling strategy considers wind but not wave. |
| **RC2.6** | 7- Could you please use labels for the curves in fig. 6? |
| **AC2.6** | The figure has been updated according to the Referee's suggestion. |
| **RC2.7** | 8- If the support substructure type is changed, how can it influence the control design? Which control parameters should be adapted? Please explain about it in the paper text. |
| **AC2.7** | The control-synthesis procedure described in the paper is applied to one platform typology, but it is valid for any platform type. We explained how the platform typology affects the wave-feedforward controller is Section 5. In detail: when a different platform is considered, the disturbance model changes, |

because forcing produced by waves depends on the platform geometry and the way waves interact with it. The FF controller responds to wave, and acts in the frequency range where most of the wave energy is. In this frequency range, the amplitude of the controller transfer function is increased if the platform is more exposed to wave excitation, that means a larger control effort is required for wave loads rejection. The platform modes are expected to change. However, these are usually outside the wave-frequency range, and have little influence on the wave-FF action. In future works, it would be nice to implement the wave-FF control strategy in FOWTs based on different platform typologies, so to reduce uncertainty about its benefits.

**RC2.8**   9- Authors' statement "Reducing rotor speed oscillations also results into a lower fatigue damage for the wind turbine shaft and tower"

Comment> Could you elaborate more on this statement in the paper text? How is rotor speed oscillation connected to fatigue damage for the wind turbine shaft and tower?

**AC2.8**   The answer to this comment is strictly connected to comment RC2.4, and, for this reason we decided to address them together in the revised paper. As said for RC2.4, we are grateful to Farid for this comment because it gives us the possibility do discuss more in the detail the coupled rotor-platform dynamics of floating turbines. Concerning RC2.8, similarly to what we commented in AC2.4, the wave-feedforward controls the blade pitch angle to reduce the aerodynamic torque fluctuations. The pitch angle modification also affects the thrust force and, consequently, the tower-FA loads.

**RC2.9**   10- Authors' statement "The proposed FB-FF controller, that keeps into account the above- mentioned movement of the LIDAR, can reduce power and rotor speed fluctuations up to 80% and tower, rotor-shaft, and blades fatigue loads of 20%, 7% and 9%, respectively"

Comment> I cannot find the results that confirm these numbers in Section Results. These numbers should match the simulation results presented in the paper.

Authors should also explain in simulation studies about those quantified results and how they are obtained.

**AC2.9**   The results mentioned in the comment pertains to a paper about LIDAR-based wind feedforward "Collective Pitch Feedforward Control of Floating Wind Turbines Using Lidar". This paper is cited in the introduction because wave-feedforward is loosely inspired to wind-feedforward control strategies. Results achieved by means of wave-feedforward are different, as the two control strategies target different disturbances.

**RC2.10**   11- Authors' statement "Wave disturbance is responsible of a large fraction of the fatigue loads experienced by a floating wind turbine"

Comment> could you please provide reference for this statement? Which failure modes are you specifically focusing on?

**AC2.10**   We agree with you that the sentence was quite broad, and some clarification was needed. First, we want to point out that the effect of wave is not limited to fatigue loads: wave provides a considerable fraction of the dynamic excitation

experience by an FOWT, and this has several effects. The effects of wave excitation are analyzed in the work of J. Jonkman "Dynamics Modeling and Loads Analysis of an Offshore Floating Wind Turbine" (2007). Wave increases rotor-speed, and generator power, oscillations because of increased platform motion. Tower shear forces and bending moments are increased as well, and the increment is proportional to the severity of waves. The suggested article "Stochastic dynamic load effect and fatigue damage analysis of drivetrains in land-based and TLP, spar and semi-submersible floating wind turbines" shows that wave is also responsible of increased fatigue damage for the drivetrain.

**RC2.11** 12- Authors' statement "The platform motion caused by waves, turns out into a variation of the apparent wind speed, which affects rotor torque, and then speed."

Comment> Could platform motions like rolling directly influence the rotor speed. Could you explain it in the text to complement the above statement?

**AC2.11** We agree appreciate this comment because it helps clarifying the assumptions we used in the analysis. 
[revised manuscript text omitted]